

# FMC² model based perception grading for dark insurgent network analysis

Ganesh Kumar Pugalendhi[1], Shanmugapriya Kumaresan[2] and Anand Paul[3]

[1] Department of Computer Science and Engineering, College of Engineering, Guindy, Anna University, Chennai, Tamilnadu, India
[2] Advanced Analytics Department, Indium Software (India) Private Ltd, Chennai, Tamilnadu, India
[3] The School of Computer Science and Engineering, Kyungpook National University, Daegu, South Korea

## ABSTRACT

The burgeoning role of social network analysis (SNA) in various fields raises complex challenges, particularly in the analysis of dark and dim networks involved in illicit activities. Existing models like the stochastic block model (SBM), exponential graph model (EGM), and latent space model (LSM) are limited in scope, often only suitable for one-mode networks. This article introduces a novel fuzzy multiple criteria multiple constraint model (FMC²) tailored for community detection in two-mode networks, which are particularly common in dark networks. The proposed method quantitatively determines the relationships between nodes based on a probabilistic measure and uses distance metrics to identify communities within the network. Moreover, the model establishes fuzzy boundaries to differentiate between the most and least influential nodes. We validate the efficacy of FMC2 using the Noordin Terrorist dataset and conduct extensive simulations to evaluate performance metrics. The results demonstrate that FMC2 not only effectively identifies communities but also ranks influential nodes within them, contributing to a nuanced understanding of complex networks. The method promises broad applicability and adaptability, particularly in intelligence and security domains where identifying influential actors within covert networks is critical.

Corresponding author
Anand Paul, paul.editor@gmail.com

# INTRODUCTION

In an interconnected world, social networks serve as intricate tapestries where diverse entities—ranging from individuals and organizations to digital domains—engage to realize shared objectives. Social network analysis (SNA) has emerged as an indispensable tool for decoding these complex webs, offering quantitative insights into the relationships and interactions that define these networks (*Everton & Roberts, 2011*). Within the taxonomy of SNA, networks can be broadly categorized into light, dim, and dark, each with distinct characteristics and implications. Light networks are transparent and open, fostering benign activities. Dim networks, while not overtly secretive, maintain a guarded interface with external organizations. Dark networks (*Milward & Raab, 2003*; *Rawat et al., 2021*), however, operate in the underbelly of society, facilitating illicit activities such as drug trafficking, money laundering, and terrorism, and thus pose challenges for comprehensive

analysis due to their concealed and dynamic nature. In the realm of intelligence analytics, social network analysis (SNA) serves as a transformative lens, fundamentally altering how analysts decipher intricate networks. While traditional networks are essentially extensions of offline social circles—comprising individuals bound by pre-existing relationships and shared activities—the landscape dramatically shifts when navigating dark networks. These clandestine networks are nebulous entities, characterized by incomplete data, ambiguous relationships, and a volatile structure, all of which defy straightforward analysis. Current methodologies in SNA predominantly rely on statistical models (*Kolaczyk, 2009*) such as the stochastic block model (SBM) (*Holland, Laskey & Leinhardt, 1983*), exponential graph model (EGM), and latent space model (LSM). While SBM pioneered community detection within networks, subsequent adaptations, notably in exponential random graph models (*Frank & Strauss, 1986*; *Robins et al., 2007*), have refined the understanding of dynamic networks. However, these models have been largely optimized for one-mode networks and falter when applied to more complex structures.

In contemporary network science, a paradigm shift is observed where networks are projected into latent spaces. This projection is primarily grounded on probabilistic assessments of inter-node relationships, further refined by distance metrics (*D'Angelo, Alfò & Fop, 2023*). Recent advancements in LSM have transcended static networks to accommodate their dynamic evolution through iterative modifications in distance measures (*Handcock, Raftery & Tantrum, 2007*; *Sewell, Chen & Etal, 2017*). However, these innovations are predominantly tailored for one-mode networks, where nodes share homogenous characteristics, thus limiting their applicability in more complex scenarios.

To address this gap, the present study introduces a novel fuzzy multiple criteria multiple constraint model (FMC$^2$), specifically designed for dissecting two-mode networks. The methodology employs probabilistic estimations of node relationships coupled with distance metrics to delineate communities within the network. Moreover, it establishes perceptual boundaries to segregate best-case and worst-case nodes within these communities, thereby identifying the most influential nodes in a hierarchical fashion. The efficacy of this perception-based grading approach has been rigorously validated using the Noordin Terrorist dataset, a publicly accessible benchmark. A series of simulations further corroborate the model's precision and robustness in both community identification and influence ranking.

The ensuing sections of this article are meticulously structured to furnish a comprehensive understanding of the research underpinning. 'Social Network Analysis' delves into the foundational principles and metrics germane to social network analysis, serving as a primer for the uninitiated. 'Model Selection' elucidates existing models, laying the groundwork for an appreciation of the limitations that this research seeks to overcome. 'Analytical Framework for Bipartite Insurgent Network Dissection' unveils the architectural blueprint of the proposed methodology, providing an aerial view of the research landscape. 'Comprehensive Execution of FMC$^2$ Model for Analyzing Insurgent Networks' offers a granular walkthrough of the implementation steps for the novel FMC2 model-based perception grading, explicating the mechanisms for community and influential node identification. 'Simulation Result' presents empirical evidence, showcasing the results

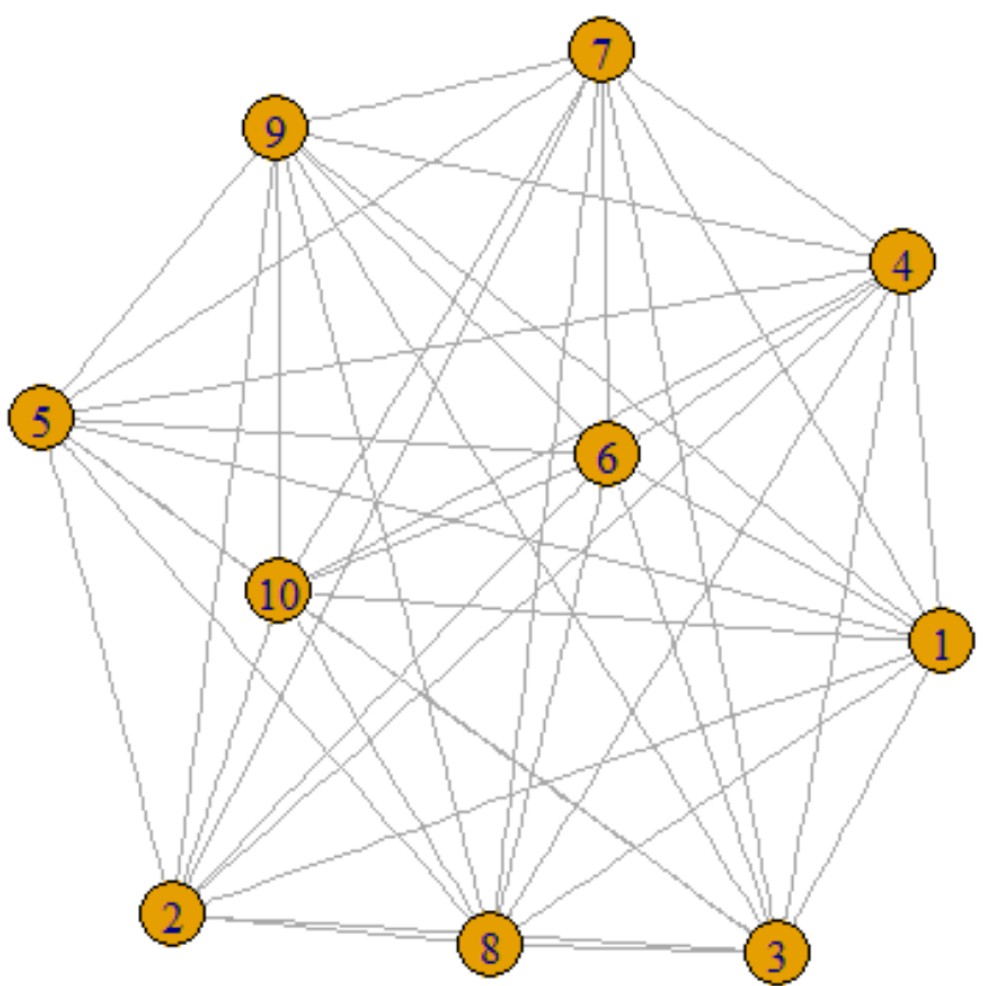

**Figure 1    One-mode projection network.**

derived from applying the proposed algorithm to real-world datasets, complemented by performance metrics. Finally, 'Conclusion and Future Work' synthesizes the overarching conclusions while charting a roadmap for future explorations in this intriguing domain.

## SOCIAL NETWORK ANALYSIS

Social network analysis (SNA) is fundamentally a graph-theoretical method, wherein vertices represent various types of actors—be they individuals, organizations, or other entities—and edges encapsulate the relationships among these vertices. Typically, networks can be categorized into one-mode and two-mode projections. In a one-mode projection, the network is formally represented as $SG = \{V, E\}$, where $V$ denotes the set of vertices and $E$ the set of edges. Such representations are commonly employed for modeling social media interactions, such as friendships on Facebook, as illustrated in Fig. 1. In these networks, nodes are labeled numerically as 1, 2, 3, up to 10, and edges are represented by lines that connect these nodes.

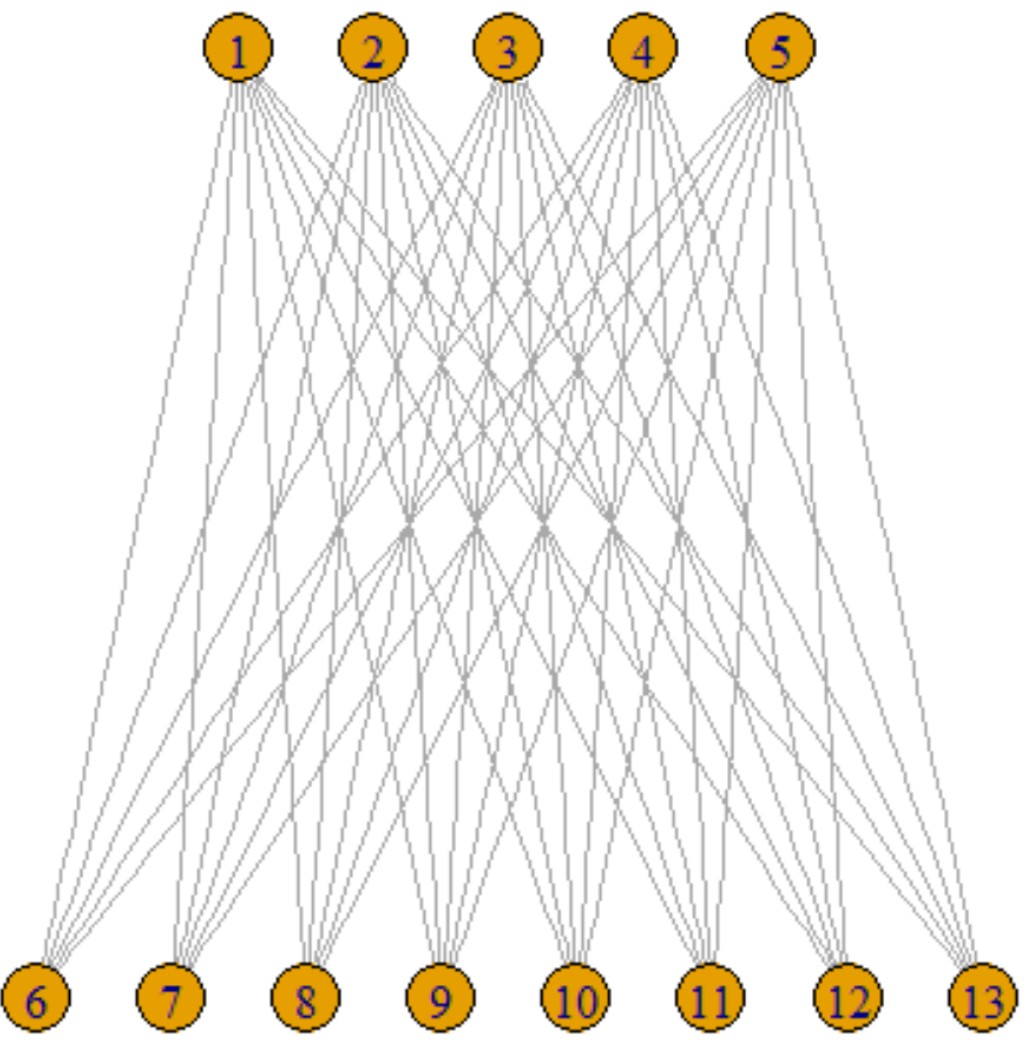

**Figure 2** Two-mode projection network.

However, one-mode projections are ill-suited for capturing the nuances of specific types of networks, such as scientific collaboration networks, where relational ties are more complex. For instance, an edge may exist between two authors if they have co-authored a paper. This leads to the concept of a two-mode, or bipartite, network. In such networks, formally represented as $BG = \{U,V,E\}$, vertices are divided into two distinct sets $U$ and $V$, with edges $E$ connecting vertices across these sets but not within them, as depicted in Fig. 2. These bipartite structures are particularly apt for modeling affiliation and bibliographic networks. This type of network is also called as affiliation network.

A multitude of metrics serve as the analytical linchpins in deciphering the complex tapestry of network structures. Among these, centrality measures—encompassing degree, closeness, betweenness, and hub scores—are particularly salient (*Hansen et al., 2019*). The subsequent sections delve into the specific centrality metrics employed in this scholarly inquiry, beginning with degree centrality. Degree centrality: regarded as a cornerstone

metric, degree centrality is fundamentally a quantitative measure that enumerates the ties a node maintains within the network. This attribute serves as an indicator of a node's relational intensity and is mathematically formalized in Eq. (1). The centrality measures used for analysis in this research work are discussed below.

### Degree centrality
Regarded as a cornerstone metric, degree centrality is fundamentally a quantitative measure that enumerates the ties a node maintains within the network. This attribute serves as an indicator of a node's relational intensity and is mathematically formalized in Eq. (1).

$$Cent_{Deg}^{i} = degree(i). \tag{1}$$

### Closeness
Closeness is the length-based measure which calculates the average shortest distance of the node. The closeness centrality of a node i is shown in Eq. (2).

$$Cent_{Clo}^{i} = \frac{1}{\sum_j dist(j,i)} \tag{2}$$

where dist(j,i) is distance between the nodes i and j.

### Betweenness
Betweenness is the quantity-based measure which calculates the number of times the node acts a bridge along the shortest path between the nodes. The nodes which are most influential will have the highest betweenness value. The betweenness centrality of a node i is shown in Eq. (3).

$$Cent_{Bet}^{i} = \sum_{i \neq j \neq k \in V} \frac{NSP_{ik}(V)}{NSP_{ik}} \tag{3}$$

where $NSP_{ik}(V)$ is the total number of shortest path from i to k.
   $NSP_{ik}$ is the number of paths through i.

### Eigen vector centrality
A sophisticated metric, Eigenvalue centrality serves as a gauge for a node's propensity to wield influence within a network structure. Perhaps the most emblematic application of Eigenvalue centrality variations is Google's PageRank algorithm, a seminal technique in search engine optimization. Mathematically, the Eigenvalue centrality of a node i is delineated in Eq. (4) as follows:

$$Cent_{Eig}^{x}(i) = \frac{1}{\alpha} \sum_{j \in N(x)} x_j \tag{4}$$

where N(x) represents the ensemble of neighboring nodes connected to $x$, and $\alpha$ is a constant scaling factor.

# MODEL SELECTION

Community detection is a sophisticated endeavor aimed at identifying nodes with elevated degrees of connectivity and assessing the extent of their influence within the network fabric. The crux of this exploration hinges on a model-based approach designed to delineate clusters or communities, anchored by their interrelational ties.

## Conventional model for two-mode networks

The two-mode network has the data in the $m \times n$ matrix with the occurrences of observation$_{ij}$, where m individuals and n events. The observation$_{ij}$ is the binary variable, where $O_{ij} = 1$, if tie exist and $O_{ij} = 0$, otherwise. Assumption of starting with X number of individuals in the community having the community extent ce = (ce$_1$, ce$_2$, ..., ce$_X$). The method will group the individuals having the common tie with the events. The conventional model is best suited only for the networks which are mutually exclusive and satisfies the following conditions as discussed by *Aitkin, Vu & Francis (2017)*.

Condition 1: $ce_x \geq 0$, for each x
Condition 2: $\sum_{x=1}^{X} ce_x = 1$.

## Multiple criteria model (MCM) framework

The multiple criteria model (MCM) serves as an intricate framework optimized for decision-making in scenarios fraught with multidimensional events and heterogenous observations, each characterized by a diverse array of attributes and parameters. Requiring domain-specific expertise, the MCM exhibits multidisciplinary versatility. Within the context of social network analysis, the MCM emerges as a robust tool, uniquely qualified to architect networks by navigating a plethora of observational choices. It adeptly identifies influential nodes, prioritizing them based on a pre-determined set of criteria. Notably, the scholarly landscape is replete with a variety of MCM paradigms, each tailored for specific decision-making applications.

ELECTRE (*Saracoglu, 2015*) and PROMETHEE P (*Velasquez & Hester, 2013*) serve as cornerstone methodologies in the realm of multi-criteria decision-making, operating on the principle of outranking. These paradigms are particularly apt for scenarios characterized by a limited set of criteria but an expansive array of alternatives. Conversely, the analytical hierarchy process (AHP) (*Rios & Duarte, 2021*) adopts a structured approach, employing pairwise comparisons to discern the most advantageous alternative. The Pugh method (*Zhu et al., 2022*), also known as the decision matrix method (DMM), takes a qualitative stance, benchmarking alternatives against a datum option. In the quantitative spectrum, the Statistical Design Institute (SDI) (*Miranda, Antunes & Gama, 2022*) computes scores for each design option. Technique for Order Preference by Similarity to an Ideal Solution (TOPSIS) (*Ogonowski, 2022*) emerges as an alternative to ELECTRE, specifically designed to pinpoint alternatives that closely approximate an ideal solution.

The quest to identify influential nodes within networks introduces a complex multi-criteria decision-making conundrum, rooted in the interplay between nodal influence weights and intricate topological attributes. This intricate dilemma serves as the intellectual impetus behind the development of our pioneering approach. Employing a fuzzy-based

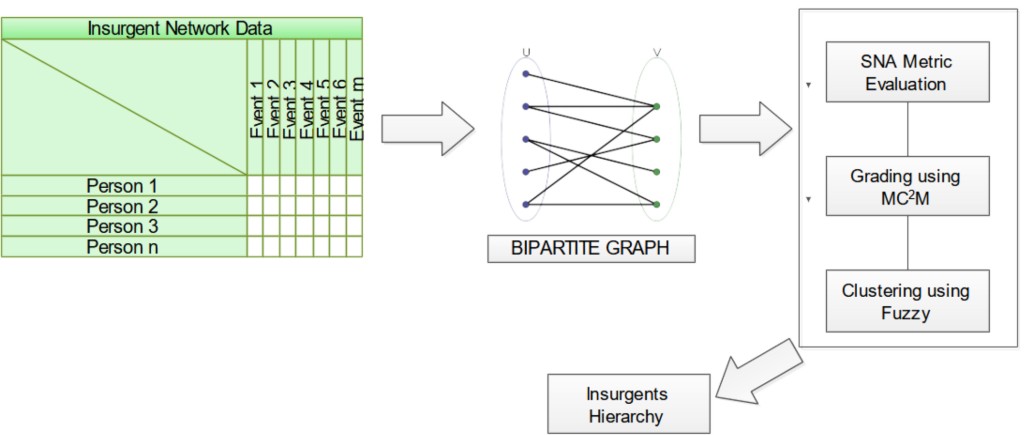

**Figure 3** Schematic diagram of proposed approach for insurgent network analysis.

multiple criteria and multiple constraint level paradigm, this methodology innovatively harnesses grading techniques and community detection algorithms for the precise identification of influential nodes.

## ANALYTICAL FRAMEWORK FOR BIPARTITE INSURGENT NETWORK DISSECTION

The methodology advanced herein focuses on the intricate analysis of two-mode insurgent networks, mathematically captured as a bipartite graph $G_b = \{T,E,R\}$, where $T$ constitutes the ensemble of terrorists, $E$ the array of events, and $R$ the relational linkage intertwining $T$ and $E$. Figure 3 elucidates the structural blueprint of this novel proposition.

The analytical ingress point is a data matrix, its rows personifying individuals purporting to be insurgents and its columns enumerating events in which these individuals partake. Matrix entries denote the individual's engagement in a specific event. The inaugural phase entails the transformation of this data matrix into a bipartite graph. Subsequently, network attributes are calculated leveraging the Jaccard coefficient, a measure elucidating the neighborhood similarity between vertices, thereby enhancing object detection fidelity. Building on this foundation, the proposed multi-criteria and multi-constraint level approach is employed for node ranking through similarity score computations. Fuzzy-based clustering techniques are applied to demarcate communities, guided by the aforementioned metrics. Culminating this analytical odyssey, individual nodes are hierarchically positioned within these communities based on the gradings computed. Finally, the ascertained influential nodes, characterized by their community-specific gradings, offer invaluable insights into the behavioral patterns of the involved insurgents.

Finally, from the set of influential nodes with the grading of nodes in community helps to track the nature of insurgents involved in the activity.

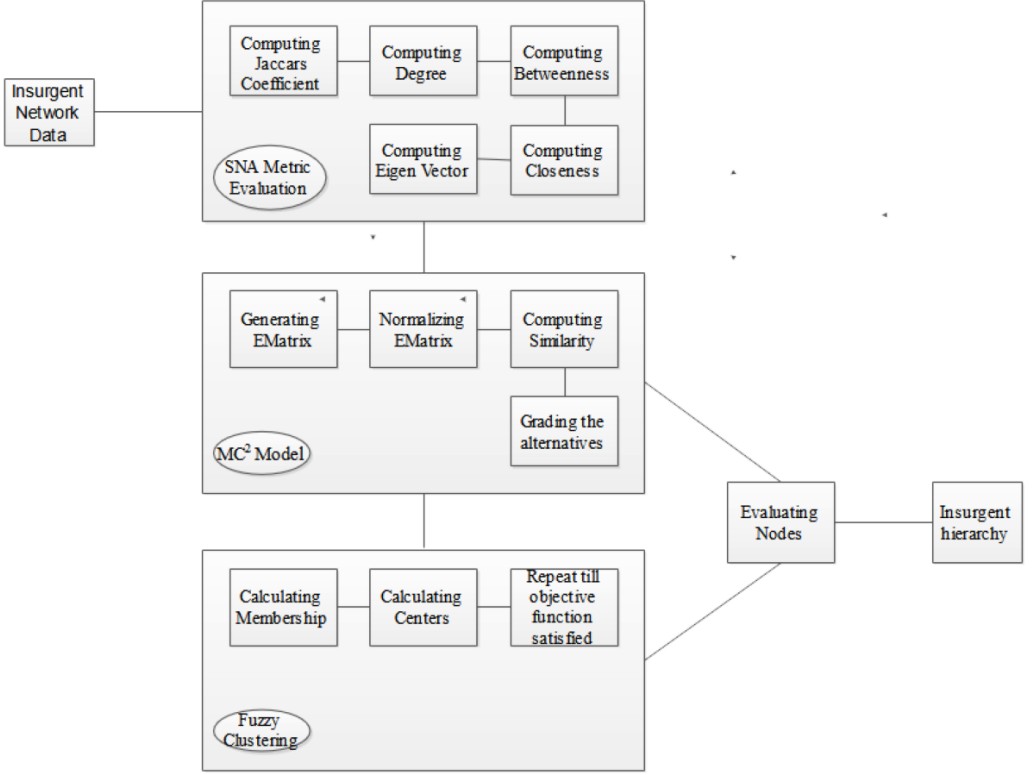

**Figure 4** **Implementation procedure of FMC2 model for insurgent network analysis.**

# COMPREHENSIVE EXECUTION OF FMC$^2$ MODEL FOR ANALYZING INSURGENT NETWORKS

This section meticulously delineates the procedural architecture underpinning the implementation of our avant-garde fuzzy multiple criteria multiple constraint model (FMC2) tailored for insurgent network scrutiny. Figure 4 provides a graphical exposition of the sequential steps implicated in the deployment of the FMC$^2$ framework.

The fulcrum of the proposed model is to engender a hierarchical taxonomy of nodes, contextualized by their event-centric involvements, thereby facilitating the demarcation of insurgents into top, middle, and bottom tiers. The analytical journey commences with the construction of a bipartite graph derived from the input network dataset. Algorithm 1, encapsulating the FMC$^2$ methodology, ingests this graph to compute the Jaccard coefficient; this is conditioned by the bipartite partitioning values, which are in turn modulated by event-specific involvements. Concomitantly, a compendium of network centrality measures—degree, closeness, betweenness, and eigenvector centrality— is calculated, each nuanced by the Jaccard coefficient. Following this, an evaluation matrix (EMatrix) is synthesized, encapsulating these computed metrics as criteria against a predefined set of alternatives. This matrix undergoes a weighted average normalization procedure, subsequent to which the extremal alternatives (best- and worst-case scenarios) are determined *via* shortest-path computations. Distances for each alternative are then

quantified, serving as the basis for similarity score calculations that ultimately grade these alternatives.

At this juncture, the influential network nodes are discernibly identified *via* the computed metrics. The algorithm uniquely accommodates multiple criteria and constraints in a unified analytical framework. Finally, EMatrix is reevaluated, and fuzzy memberships $\mu_i$ and fuzzy centers are iteratively computed until the objective function converges, subject to $n$ observations and $k$ clusters. Additionally, $dsim(i,j)$ signifies the dissimilarity measure between observations, and $e$ represents the membership exponent

Algorithm 1: FMC$^2$ Method

Input:

Bg- bipartite graph

Output:

Comm –Set of Clusters

Comm_infl –Community wise Influential nodes in hierarchical position

Method:

procedure FMC$^2$Approach (Bg)

{

Bp =bipartite_partition_binary(Bg)

for each vertex in Bp{

set $Bp_{11} = x$, $Bp_{10} = y$, $Bp_{01} = z$

$JS[i] = \frac{x}{x+y+z}$

if($JS[i] > 0.95$)

    pJaccard[i] = 1

else

    pJaccard[j] = 0

diagonal of pJaccard = 0

$Cent_{Deg}^{i} = degree(i)$

$Cent_{Clo}^{i} = \frac{1}{\sum_j dist(j,i)}$

$Cent_{Bet}^{i} = \sum_{i \neq j \neq k \in V} \frac{NSP_{ik}(V)}{NSP_{ik}}$

$Cent_{Eig}^{x}(i) = \frac{1}{\alpha}\sum_{j \in N(x)} x_j$

}

Generate EMatrix$_{nxm}$

Calculate EMatrix_Norm$_{nxm}$ for each choice $n_{ij} = \frac{EMatrix_{ij}}{\sqrt{\sum EMatrix_{ij}^2}}$

$W = (w_{ij})_{nxm} = (a_j n_{ij})_{nxm}$

$S_w = distance$

$S_b = distance$

$iw = \frac{S_w}{S_w + S_b}$

Comm_infl = Grade ($iw$

Comm = $F_{min} = \sum_{c=1}^{k} \frac{\sum_{i,j=1}^{n} \mu_{ic}^e \mu_{jc}^e dsim(i,j)}{2\sum_{j=1}^{n} \mu_{jc}^e}$

}

## SIMULATION RESULT

In this section, simulation carried out using 'R' (*Murdoch, 2017*) for the proposed work is discussed and the results are reported.

### Dataset utilization: analyzing noordin top's terrorist network

In the present study, we employ the Noordin Top Terrorist Network dataset (*International Crisis Group, 2006*), a publicly accessible corpus curated by the International Crisis Group. This dataset encapsulates data on 79 individuals implicated in extremist activities, with a specific focus on operations staged in Jakarta and Bali, Indonesia, in the year 2011. The dataset is comprehensive, encompassing a total of 568 distinct events, some of which wield significant influence on the affiliative dynamics within extremist organizations. The scope of affiliations scrutinized in our analysis spans a diverse range of institutions— educational establishments (schools and colleges), commercial enterprises, and religious organizations—as well as an array of relational dimensions such as classmate interactions, familial ties, friendships, co-religious affiliations, and other logistical support networks involved in training, terrorist activities, and strategic assemblies..

### SNA metric evaluation—quantifying node attributes in the network

Table 1 delineates a suite of computational metrics—namely, degree, betweenness, closeness, and eigenvector centrality—applied to the 79 nodes constituting the network under study. The "degree" serves as an indicator of a node's connectivity within the network, quantifying the number of edges emanating from or converging to it. "Betweenness" furnishes insights into a node's role as a connective bottleneck or gateway, bridging disparate clusters within the network. The "closeness" metric elucidates the extent to which a node is intimately connected to others in the network, essentially serving as a measure of its reachability. Lastly, "eigenvector centrality" offers a nuanced understanding of a node's influence, taking into account not merely the quantity but the quality of its connections.

### FMC$^2$ based perception grading method

The similarity score computed by applying the proposed FMC$^2$ method and the grade based on the computed score are depicted in the Table 2.

The scores, delineated in Fig. 5, manifest as fuzzy-based values in the interval $[0,1][0,1]$, serving to classify the nodes on a continuum from least to most influential. The $x$-axis enumerates the Node IDs, while the $y$-axis represents the corresponding grade values. Notably, Node 59 emerges as an apex entity, registering a grade value of 1. Contrary to initial impressions, it should be emphasized that within the context of this network, a lower grade value paradoxically indicates heightened influence as opposed to a higher grade value.

### Fuzzy clustering

Fuzzy clustering, leveraging the output from the FMC$^2$ model, initiates with a bifurcated cluster framework and iteratively expands to three clusters. Table 3 encapsulates the nuanced metrics: **Dunn_Coeff**, representing the partition coefficient of the clustering, is

**Table 1  Summary of centrality measures of the noordin data set.**

| Person ID | Degree | Betweeness | Closeness | Eigen centrality |
|---|---|---|---|---|
| 1 | 222 | 272.8423 | 0.0029 | 0.8294 |
| 2 | 212 | 272.5252 | 0.0028 | 0.7642 |
| 3 | 220 | 272.7874 | 0.0028 | 0.8136 |
| 4 | 219 | 272.6826 | 0.0028 | 0.8093 |
| 5 | 243 | 275.1958 | 0.0030 | 0.9489 |
| 6 | 216 | 272.6378 | 0.0028 | 0.7870 |
| 7 | 210 | 272.5108 | 0.0028 | 0.7517 |
| 8 | 214 | 272.5477 | 0.0028 | 0.7767 |
| 9 | 214 | 272.5717 | 0.0028 | 0.7803 |
| 10 | 209 | 272.5063 | 0.0028 | 0.7453 |
| 11 | 214 | 272.5676 | 0.0028 | 0.7754 |
| 12 | 209 | 272.5063 | 0.0028 | 0.7460 |
| 13 | 220 | 272.7990 | 0.0028 | 0.8128 |
| 14 | 214 | 272.5717 | 0.0028 | 0.7803 |
| 15 | 234 | 273.9582 | 0.0030 | 0.8983 |
| 16 | 217 | 272.6415 | 0.0028 | 0.7992 |
| 17 | 209 | 272.5063 | 0.0028 | 0.7449 |
| 18 | 217 | 272.6627 | 0.0028 | 0.7970 |
| 19 | 214 | 272.5647 | 0.0028 | 0.7792 |
| 20 | 215 | 272.5992 | 0.0028 | 0.7864 |
| 21 | 214 | 272.5635 | 0.0028 | 0.7800 |
| 22 | 216 | 272.6213 | 0.0028 | 0.7889 |
| 23 | 244 | 275.4381 | 0.0030 | 0.9532 |
| 24 | 232 | 273.7902 | 0.0029 | 0.8823 |
| 25 | 209 | 272.5063 | 0.0028 | 0.7460 |
| 26 | 216 | 272.6120 | 0.0028 | 0.7911 |
| 27 | 211 | 272.5188 | 0.0028 | 0.7603 |
| 28 | 212 | 272.5249 | 0.0028 | 0.7645 |
| 29 | 213 | 272.5467 | 0.0028 | 0.7727 |
| 30 | 218 | 272.7110 | 0.0028 | 0.8044 |
| 31 | 226 | 273.1580 | 0.0029 | 0.8485 |
| 32 | 213 | 272.5510 | 0.0028 | 0.7726 |
| 33 | 210 | 272.5108 | 0.0028 | 0.7517 |
| 34 | 212 | 272.5339 | 0.0028 | 0.7648 |
| 35 | 214 | 272.5728 | 0.0028 | 0.7796 |
| 36 | 220 | 272.8028 | 0.0028 | 0.8170 |
| 37 | 216 | 272.5846 | 0.0028 | 0.7910 |
| 38 | 209 | 272.5063 | 0.0028 | 0.7452 |
| 39 | 224 | 272.9949 | 0.0029 | 0.8361 |
| 40 | 209 | 272.5063 | 0.0028 | 0.7446 |
| 41 | 216 | 272.6311 | 0.0028 | 0.7927 |
| 42 | 217 | 272.6648 | 0.0028 | 0.7983 |

**Table 1** (*continued*)

| Person ID | Degree | Betweeness | Closeness | Eigen centrality |
|---|---|---|---|---|
| 43 | 211 | 272.5153 | 0.0028 | 0.7591 |
| 44 | 214 | 272.5754 | 0.0028 | 0.7763 |
| 45 | 230 | 273.5313 | 0.0029 | 0.8718 |
| 46 | 237 | 274.4094 | 0.0030 | 0.9111 |
| 47 | 212 | 272.5317 | 0.0028 | 0.7671 |
| 48 | 210 | 272.5105 | 0.0028 | 0.7531 |
| 49 | 214 | 272.5717 | 0.0028 | 0.7803 |
| 50 | 225 | 273.0542 | 0.0029 | 0.8451 |
| 51 | 214 | 272.5533 | 0.0028 | 0.7769 |
| 52 | 219 | 272.7466 | 0.0028 | 0.8074 |
| 53 | 212 | 272.5198 | 0.0028 | 0.7655 |
| 54 | 211 | 272.5193 | 0.0028 | 0.7598 |
| 55 | 217 | 272.6642 | 0.0028 | 0.7987 |
| 56 | 228 | 273.3755 | 0.0029 | 0.8623 |
| 57 | 223 | 272.9501 | 0.0029 | 0.8346 |
| 58 | 212 | 272.5322 | 0.0028 | 0.7666 |
| 59 | 252 | 276.8691 | 0.0031 | 1.0000 |
| 60 | 216 | 272.5995 | 0.0028 | 0.7902 |
| 61 | 211 | 272.5198 | 0.0028 | 0.7586 |
| 62 | 212 | 272.5317 | 0.0028 | 0.7671 |
| 63 | 224 | 273.0142 | 0.0029 | 0.8393 |
| 64 | 227 | 273.2198 | 0.0029 | 0.8546 |
| 65 | 215 | 272.5992 | 0.0028 | 0.7864 |
| 66 | 213 | 272.5506 | 0.0028 | 0.7729 |
| 67 | 210 | 272.5063 | 0.0028 | 0.7527 |
| 68 | 215 | 272.6024 | 0.0028 | 0.7832 |
| 69 | 209 | 272.5063 | 0.0028 | 0.7460 |
| 70 | 224 | 273.0291 | 0.0029 | 0.8370 |
| 71 | 213 | 272.5520 | 0.0028 | 0.7707 |
| 72 | 208 | 272.5063 | 0.0027 | 0.7384 |
| 73 | 214 | 272.5753 | 0.0028 | 0.7762 |
| 74 | 213 | 272.5411 | 0.0028 | 0.7734 |
| 75 | 209 | 272.5063 | 0.0028 | 0.7451 |
| 76 | 212 | 272.5322 | 0.0028 | 0.7666 |
| 77 | 211 | 272.5154 | 0.0028 | 0.7580 |
| 78 | 210 | 272.5109 | 0.0028 | 0.7516 |
| 79 | 210 | 272.5110 | 0.0028 | 0.7513 |

**Table 2  Summary of similarity score and grade evaluated for Noordin network.**

| ID | Score | Grade | ID | Score | Grade | ID | Score | Grade |
|----|-------|-------|----|-------|-------|----|-------|-------|
| 1  | 0.3343 | 16   | 28 | 0.0956 | 58   | 55 | 0.2194 | 24   |
| 2  | 0.0948 | 59   | 29 | 0.1238 | 49   | 56 | 0.4631 | 8    |
| 3  | 0.2793 | 18   | 30 | 0.2412 | 22   | 57 | 0.3549 | 15   |
| 4  | 0.2609 | 20   | 31 | 0.4129 | 10   | 58 | 0.1010 | 54.5 |
| 5  | 0.7984 | 3    | 32 | 0.1235 | 50   | 59 | 1.0000 | 1    |
| 6  | 0.1824 | 32   | 33 | 0.0483 | 67.5 | 60 | 0.1904 | 30   |
| 7  | 0.0483 | 67.5 | 34 | 0.0963 | 57   | 61 | 0.0732 | 63   |
| 8  | 0.1412 | 43   | 35 | 0.1486 | 40   | 62 | 0.1023 | 52.5 |
| 9  | 0.1505 | 37   | 36 | 0.2880 | 17   | 63 | 0.3746 | 12   |
| 10 | 0.0248 | 74   | 37 | 0.1924 | 29   | 64 | 0.4361 | 9    |
| 11 | 0.1380 | 46   | 38 | 0.0245 | 75   | 65 | 0.1734 | 33.5 |
| 12 | 0.0265 | 72   | 39 | 0.3666 | 14   | 66 | 0.1245 | 48   |
| 13 | 0.2775 | 19   | 40 | 0.0229 | 78   | 67 | 0.0510 | 66   |
| 14 | 0.1505 | 37   | 41 | 0.1968 | 27   | 68 | 0.1652 | 35   |
| 15 | 0.6001 | 5    | 42 | 0.2182 | 25   | 69 | 0.0265 | 72   |
| 16 | 0.2206 | 23   | 43 | 0.0745 | 62   | 70 | 0.3689 | 13   |
| 17 | 0.0238 | 77   | 44 | 0.1404 | 44   | 71 | 0.1188 | 51   |
| 18 | 0.2150 | 26   | 45 | 0.5024 | 7    | 72 | 0.0000 | 79   |
| 19 | 0.1478 | 41   | 46 | 0.6560 | 4    | 73 | 0.1400 | 45   |
| 20 | 0.1734 | 33.5 | 47 | 0.1023 | 52.5 | 74 | 0.1256 | 47   |
| 21 | 0.1498 | 39   | 48 | 0.0521 | 65   | 75 | 0.0242 | 76   |
| 22 | 0.1869 | 31   | 49 | 0.1505 | 37   | 76 | 0.1010 | 54.5 |
| 23 | 0.8176 | 2    | 50 | 0.3967 | 11   | 77 | 0.0716 | 64   |
| 24 | 0.5446 | 6    | 51 | 0.1418 | 42   | 78 | 0.0482 | 69   |
| 25 | 0.0265 | 72   | 52 | 0.2563 | 21   | 79 | 0.0473 | 70   |
| 26 | 0.1926 | 28   | 53 | 0.0981 | 56   |    |        |      |
| 27 | 0.0776 | 60   | 54 | 0.0763 | 61   |    |        |      |

normalized and designated as ***Normalized***. ***Obj_Func*** signifies the nadir of the objective function achieved through relative convergence tolerance, while the iterative count and average cluster width are also detailed.

Figure 6 delineates three distinct clusters, boasting cluster widths of 0.2555, 0.62588, and 0.53196, respectively. The constituent entities in these clusters are 16 for Cluster 1, 33 for Cluster 2, and 30 for Cluster 3.

The ramifications of this clustering paradigm manifest as overlapping zones within the clusters, introducing what we term as 'confusing actors' in network analytics. These actors present a deceptive semblance of influence owing to the specific nature of their network ties, thereby complicating the analytical landscape.

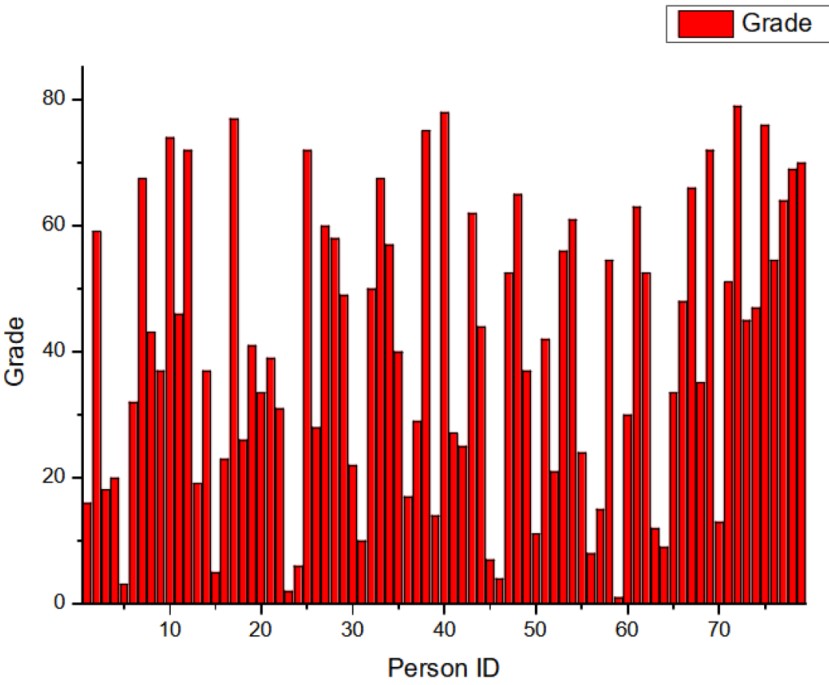

**Figure 5** Grade distribution.

**Table 3** Summary of fuzzy clustering.

| Number of cluster | Dunn_Coeff | Normalized | Obj Func | Iterations | Average width |
|---|---|---|---|---|---|
| 2 | 0.99185 | 0.98371 | 184.9 | 30 | 0.704906 |
| 3 | 0.98789 | 0.98184 | 133.9 | 22 | 0.5152 |

Figure 7 offers a visual representation of node clustering, wherein nodes are spatially organized based on their computed grades, set against their respective clusters.

Table 4 promulgates the hierarchical ranking of nodes within these clusters, serving as a testament to the efficacy of the proposed method.

Table 4 illuminates a structured hierarchy of insurgent actors, providing an intricate mapping of their relative import within terrorist activities in Indonesia. Arising from the application of our avant-garde methodology, we delineate the terrorist network into three concentric tiers:

(a) The central nexus comprises the preeminent figures who wield unparalleled authority and orchestrate the attacks with dictatorial command.

(b) The intermediate echelon consists of influential personas functioning as conduits between the central authority and the periphery. They are tethered to the central figures *via* operational and relational channels.

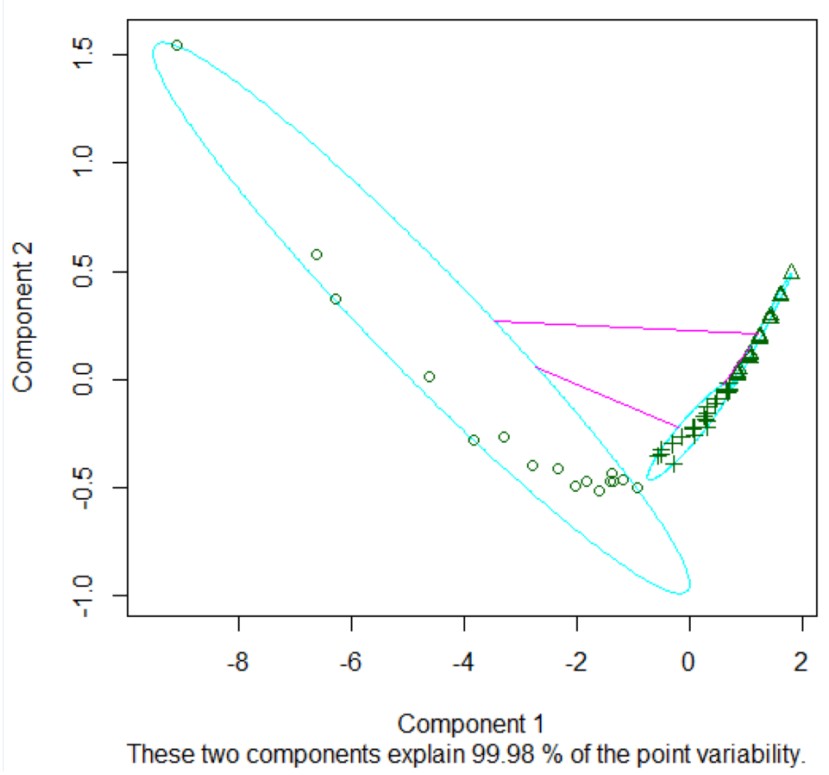

These two components explain 99.98 % of the point variability.

**Figure 6** Clusters of $k = 3$.

(c) The peripheral cadre, while less influential, serve as the logistical backbone to the operation and lack direct affiliations with the central authority.

Figure 8 graphically encapsulates this hierarchical stratification, offering a visually compelling elucidation of the intricate layers of insurgent influence.

## Performance analysis: empirical evaluation on the Noordin top terrorist dataset

Table 5 furnishes a rigorous quantitative assessment of our proposed algorithm's efficacy in identifying influential nodes and effecting data clustering within the Noordin Top Terrorist Network dataset. The performance indicators encapsulated therein include sensitivity, specificity, positive predictive value (PPV), and negative predictive value (NPV).

*Sensitivity*: This metric quantifies the algorithm's proficiency in accurately pinpointing nodes of genuine influence within the network.

*Specificity*: Complementary to sensitivity, this measure assesses the algorithm's capability to correctly identify nodes that are not influential, thereby mitigating the risk of false positives. Positive predictive value (PPV) & negative predictive value (NPV): These values offer an enhanced understanding of the algorithm's predictive precision, calibrated against the overall prevalence of influential nodes within the network.

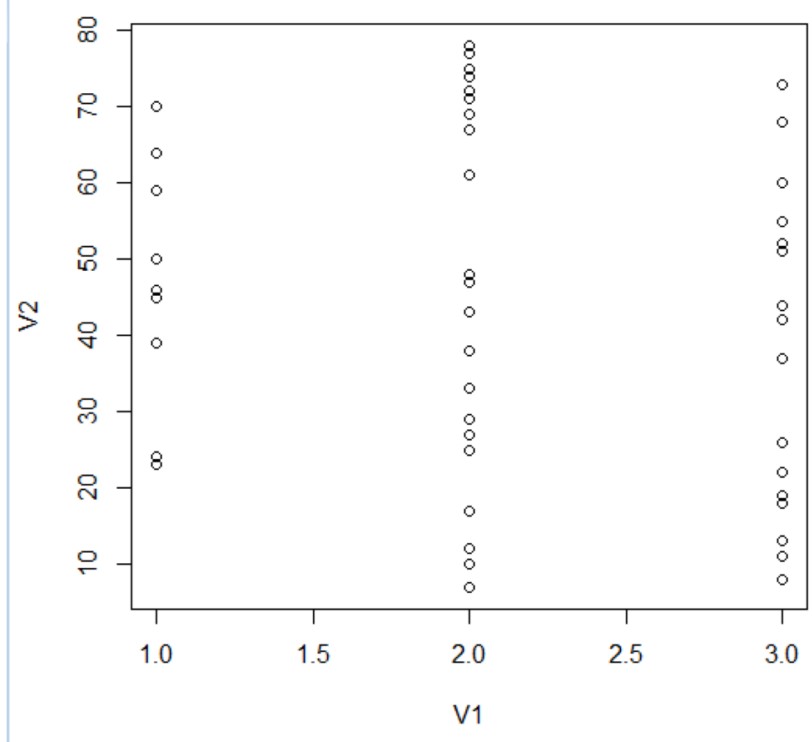

Figure 7  Structuring of cluster.

# CONCLUSION AND FUTURE WORK

In this article, we have introduced a groundbreaking methodology leveraging a fusion of fuzzy-based multi-criteria and multi-constraint mechanisms for grading nodes, thereby enabling the identification of influential actors in insurgent networks. Employing a suite of social network analysis metrics—namely, degree, betweenness, closeness, and eigenvector centrality—our approach harmonizes these conventional measures with fuzzy membership and modularity metrics to formulate a sophisticated community detection model. This facilitates the nuanced clustering of nodes within the complex structure of insurgent networks.

Our fuzzy multiple criteria multiple constraint level approach (FMC2) serves as the computational backbone for this undertaking. By computing similarity measures through a distance-based MC2 method, the FMC2 framework yields graded values that not only identify but also hierarchically classify influential nodes within the network. Furthermore, our methodology incorporates fuzzy boundaries in centrality-based measures, enhancing the precision of the resultant clusters.

This innovative approach is explicitly tailored for two-mode insurgent networks, accommodating multiple criteria and constraints concurrently for a more comprehensive analysis. However, its applicability is not confined to this specific type

**Table 4 Summary of grade for nodes in clusters.**

| Cluster 1 | | Cluster 2 | | Cluster 3 | |
|---|---|---|---|---|---|
| Node | Grade | Node | Grade | Node | Grade |
| 59 | 1 | 13 | 10 | 74 | 26 |
| 23 | 2 | 52 | 11 | 29 | 27 |
| 46 | 3 | 55 | 12 | 71 | 28 |
| 24 | 4 | 42 | 13 | 47 | 29 |
| 45 | 5 | 18 | 14 | 27 | 30 |
| 64 | 6 | 26 | 15 | 43 | 31 |
| 50 | 7 | 37 | 16 | 61 | 32 |
| 70 | 8 | 60 | 17 | 77 | 33 |
| 39 | 9 | 22 | 18 | 48 | 34 |
| | | 68 | 19 | 67 | 35 |
| | | 19 | 20 | 7 | 36 |
| | | 51 | 21 | 33 | 37 |
| | | 8 | 22 | 78 | 38 |
| | | 44 | 23 | 12 | 39 |
| | | 73 | 24 | 25 | 40 |
| | | 11 | 25 | 69 | 41 |
| | | | | 10 | 42 |
| | | | | 38 | 43 |
| | | | | 75 | 44 |
| | | | | 17 | 45 |
| | | | | 72 | 46 |

**Table 5 Performance of the categorization of persons in the Noordin network.**

| | | Data set | | Sensitivity | Specificity | Positive predicted value | Negative predicted value |
|---|---|---|---|---|---|---|---|
| | | Terrorist | Not terrorist | | | | |
| Implementation result | Terrorist | 46 | 4 | 0.9787 | 0.875 | 0.92 | 0.92 |
| | Not terrorist | 1 | 28 | | | | |

of network; it can be readily extended to other two-mode network structures. Looking ahead, future variations of this work could integrate machine learning techniques into our existing framework, offering yet another dimension of analytical depth.

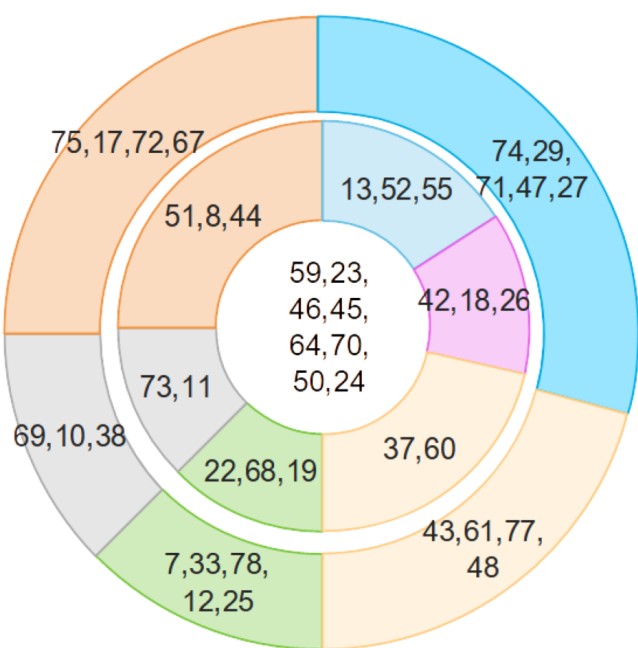

**Figure 8** Hierarchies of insurgents.

### Funding

Anand Paul was supported by the BK21 FOUR Project (AI-driven Convergence Software Education Research Program) through the Ministry of Education, School of Computer Science and Engineering, Kyungpook National University, South Korea, under Grant 4199990214394. This work was also supported by the National Research Foundation of Korea (NRF) grants funded by the Korean government, Grant Number: 2020R1A2C1012196. The funders had no role in study design, data collection and analysis, decision to publish, or preparation of the manuscript.

### Grant Disclosures

The following grant information was disclosed by the authors:
The BK21 FOUR Project (AI-driven Convergence Software Education Research Program) through the Ministry of Education, School of Computer Science and Engineering, Kyungpook National University, South Korea: 4199990214394.
The National Research Foundation of Korea (NRF) grants funded by the Korean government: 2020R1A2C1012196.

### Competing Interests

Anand Paul is an Academic Editor for PeerJ.
Shanmugapriya Kumaresan is an employee of Indum Software (India) Private Limited.

## Author Contributions

- Ganesh Kumar Pugalendhi conceived and designed the experiments, performed the experiments, analyzed the data, prepared figures and/or tables, authored or reviewed drafts of the article, formulating the concept alongside with Shanmugapriya Kumeresan, and approved the final draft.
- Shanmugapriya Kumaresan conceived and designed the experiments, performed the experiments, analyzed the data, performed the computation work, prepared figures and/or tables, and approved the final draft.
- Anand Paul conceived and designed the experiments, authored or reviewed drafts of the article, resource allocation and Research Discussion, and approved the final draft.

## Data Availability

The code and raw data are available in the Supplemental Files.

## Supplemental Information

Supplemental information for this article can be found online at http://dx.doi.org/10.7717/peerj-cs.1644#supplemental-information.

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
