# Peer review of "FMC2 model based perception grading for dark insurgent network analysis"

_PeerJ Computer Science, doi:10.7717/peerj-cs.1644_

## Round 0.1 · original submission · Major Revisions

This paper proposes a hybrid approach of multiple criteria multiple constraint level decision making followed by perception-based grading scheme to analyze the Noordin Insurgent network to identify the key influential nodes. However, some reviewer comments should be addressed.

Reviewer 1 ·

Basic reporting

no comment

Experimental design

no comment'

Validity of the findings

no comment

Additional comments

This paper studies a hybrid approach of multiple criteria multiple constraint level decision making followed by perception-based grading scheme to analyze the Noordin Insurgent network to identify the key influential nodes. This paper seems interesting, however, the authors are required to address the following questions:


The language of this paper needs further polishing.

Some recent relevant references can be added to enrich the research background of this paper.

The difficulties encountered in this article need to be further clarified, and how did the author solve these difficulties?

Adding comparisons between this paper and other recent research in related fields can increase the impact of this paper.


The innovation points of this article need to be further clarified, especially comparing the latest research in related fields.

Reviewer 2 ·

Basic reporting

The paper is interesting and has a great idea. However, the article is not well written and requires thorough editing. Some examples where editing is required include lines Line 21-22 make it clear – e.g., “the study applies these methods to identify the key influential nodes from the Noordin….”, Line 31, Line 38, Line 59 – 60 unclear “the proposed method calculates the probability of relationship between the nodes in the network and based on distance measure evaluated the communities are grouped.” Line 166 is unclear, Line 81 is it face book or Facebook?

The introduction and background need more detail. For instance, in line 35, you should explain and cite what you mean by light networks and dim networks. Line 41 is unclear, explain how SNA is intelligent, what is traditional SNA? In line 57, is it FMC2M or FMC2 as written in the title? Be consistent.

Within the literature review section (2. Social Network Analysis), provide citations. Most of the references are from Wikipedia, e.g. [11, 13], etc. There is nothing wrong with consulting Wikipedia but for a research article, you should aim to cite an original research source. For instance, for centrality measures or SNA in general, you may cite:
- R. Grassi, F. Calderoni, M. Bianchi, A. Torriero (2019). Betweenness to assess leaders in criminal networks: New evidence using the dual projection approach. Social Networks 56 (2019) 23–32.
- Freeman, L. C. (1978/79). Centrality in Social Networks Conceptual Clarification. Social Networks 1: 215–239.
- Scott, J. (2000). Social Network Analysis. Newbury Park, California: Sage.
- Borgatti, S. P., Everett, M. G., & Johnson, J. C. (2018). Analyzing social networks (2nd ed.). Los Angeles, CA: Sage
- Freeman, L., Borgatti, S., White, D., 1991. Centrality in valued graphs: a measure of betweenness based on network flow. Soc. Netw. 141–154.
- Newman, M., 2010. Networks: An Introduction. Oxford University Press, New York.

Line 324 of your reference needs editing, delete the sentence "your references here".

The article has 10 figures and 5 tables, some of them do not add much information to the study. For instance, Figures 7 and 8 are similar, select one and explain the results.

Experimental design

The authors proposed a Fuzzy Multiple Criteria Multiple Constraint Model, but it is unclear why the other models (lines 138-159) differ from the novel method of fuzzy-based multiple criteria and multiple constraints. To fill the gap/add to the literature, I suggest expanding on this section to explain how and why the novel fuzzy model is different or better than the other models available for this problem.

I also suggest expanding the methods with sufficient detail. For example, in line 126, explain what you mean by community detection. Line 146 Figure 3 is irrelevant, not adding any new information to your writing. Line 170, explain and cite why you used the Jaccard coefficient. Line 208 – 212, which ties/events were used for the analysis? explain.

Validity of the findings

The data and R script code provided are robust, statistically sound, and well-documented. However, some detailed explanations of the results are missing.
For example, in lines 215 – 216 and Table 1, what do the results of centrality measures mean? Explain.
Lines 219 – 220 describe the results of Table 2. What do the score and grade mean? for example, what do the score and grade mean for node ID 1, etc?
Line 218 – 224 repetition
Lines 232 – 234, explain the results of the fuzzy clustering. What do the numbers mean? Line 241 explain what you mean by confusing actors in the network analysis. Line 252 is unclear, lines 260 – 263 explain the results of the performance analysis.

Additional comments

The research is interesting and relevant to the field. There is a potential for a great research article. I hope the authors will consider editing and revising the article by providing detailed information and citation regarding their proposed method and results for identifying key influential nodes and communities from social network data.

---

## Round 0.2 · accepted · Accept

The paper has addressed all issues.

Reviewer 1 ·

Basic reporting

Clear.

Experimental design

Original.

Validity of the findings

Impact.

Additional comments

The authors responded nicely to all my questions without further comment.